# CSTrack: Enhancing RGB-X Tracking via Compact Spatiotemporal Features

Xiaokun Feng [* 1 2]   Dailing Zhang [* 1 2]   Shiyu Hu [3]   Xuchen Li [1 2]
Meiqi Wu [4]   Jing Zhang [2]   Xiaotang Chen [2]   Kaiqi Huang [1 2]

## Abstract

Effectively modeling and utilizing spatiotemporal features from RGB and other modalities (*e.g.*, depth, thermal, and event data, denoted as X) is the core of RGB-X tracker design. Existing methods often employ two parallel branches to separately process the RGB and X input streams, requiring the model to simultaneously handle two dispersed feature spaces, which complicates both the model structure and computation process. More critically, intra-modality spatial modeling within each dispersed space incurs substantial computational overhead, limiting resources for inter-modality spatial modeling and temporal modeling. To address this, we propose a novel tracker, *CSTrack*, which focuses on modeling *C*ompact *S*patiotemporal features to achieve simple yet effective tracking. Specifically, we first introduce an innovative *Spatial Compact Module* that integrates the RGB-X dual input streams into a compact spatial feature, enabling thorough intra- and inter-modality spatial modeling. Additionally, we design an efficient *Temporal Compact Module* that compactly represents temporal features by constructing the refined target distribution heatmap. Extensive experiments validate the effectiveness of our compact spatiotemporal modeling method, with CSTrack achieving new SOTA results on mainstream RGB-X benchmarks. The code and models will be released at: https://github.com/XiaokunFeng/CSTrack.

[*]Equal contribution [1]School of Artificial Intelligence, University of Chinese Academy of Sciences, Beijing, China [2]The Key Laboratory of Cognition and Decision Intelligence for Complex Systems, Institute of Automation, Chinese Academy of Sciences, Beijing, China [3]School of Physical and Mathematical Sciences, Nanyang Technological University, Singapore [4]School of Computer Science and Technology, University of Chinese Academy of Sciences, Beijing, China. Correspondence to: Kaiqi Huang <kaiqi.huang@nlpr.ia.ac.cn>.

*Proceedings of the 42nd International Conference on Machine Learning*, Vancouver, Canada. PMLR 267, 2025. Copyright 2025 by the author(s).

## 1. Introduction

As a fundamental visual task, object tracking (Yilmaz et al., 2006) aims to localize a target object within a video sequence based on its initial position. To achieve robust tracking in complex and corner cases, such as occlusions (Stadler & Beyerer, 2021), low visibility (Zhang et al., 2022a), and fast-moving objects (Tang et al., 2022), leveraging the complementary advantages of RGB and other modalities (*e.g.*, depth, thermal, and event data, collectively denoted as X) has emerged as a promising approach. Recently, several specialized benchmarks (Yan et al., 2021b; Li et al., 2021; Wang et al., 2023) have been introduced, along with a growing body of outstanding RGB-X trackers (Zhu et al., 2023a; Hong et al., 2024; Hou et al., 2024; Wu et al., 2024; Chen et al., 2025; Hu et al., 2025).

Auxiliary modality X enhances the potential for robust tracking but requires the tracker to handle RGB-X dual input streams simultaneously. Effective modeling and utilization of RGB-X spatiotemporal features are central to tracker design (Zhang et al., 2024b). Existing trackers typically adopt two parallel branches to process the RGB and X modalities separately, each retaining its own feature space (Hou et al., 2024). Specifically, they often initialize the RGB branch using the backbone of a pre-trained RGB tracker (Ye et al., 2022) and design dedicated architectures for the X branch. For example, ViPT (Zhu et al., 2023a) and One-Tracker (Hong et al., 2024) leverage the paradigm of prompt learning (Lester et al., 2021) to design the X branch, where the X modality is processed as prompts for utilization. In contrast, SDSTrack (Hou et al., 2024) and BAT (Cao et al., 2024) employ symmetrical architectures, using the similar RGB-tracker backbone to process the X modality.

Although achieving certain effectiveness, these methods require handling two modality-specific dispersed feature spaces simultaneously. This necessitates performing intra-modality spatial modeling between search images and template cues within each modality while also considering feature interactions across modalities, which complicates both the model structure and computation process. More critically, intra-modality modeling within each dispersed space incurs significant computational overhead, leaving insufficient room for other spatiotemporal feature modeling, result-

ing in the following limitations. **1) For spatial modeling**: There exists a severe imbalance between the intra- and inter-modality modeling. For instance, in BAT (Cao et al., 2024), the parameter proportions for intra- and inter-modality computations are 92.0% and 0.3%, respectively. Insufficient interaction across modalities limits the potential to fully exploit the complementary advantages of multimodal information. **2) For temporal modeling**: Most existing RGB-X trackers (Hong et al., 2024; Hou et al., 2024; Wu et al., 2024) either lack temporal modeling or rely solely on sparse temporal cues, such as dynamic templates (Wang et al., 2024a; Sun et al., 2024), which restricts their robustness when facing challenging scenarios.

Given the limitations of handling two dispersed spaces and the highly aligned and overlapping spatial semantic information between RGB and X images (Yang et al., 2022; Wu et al., 2024), a natural question arises: **Is it necessary to always maintain these two dispersed spaces?** The Attenuation Theory (Treisman, 1964) suggests that, when faced with multiple sources of information, the human brain follows a coarse-to-fine processing approach. It integrates the information into a small set of key elements before further processing (Zhao et al., 2024). Motivated by this, we propose a novel tracker, *CSTrack*, which emphasizes modeling and utilizing the *C*ompact *S*patiotemporal feature. By employing a single-branch compact spatiotemporal feature, we can overcome the limitations of dual dispersed spaces, enabling more effective multimodal spatiotemporal modeling.

Specifically, CSTrack consists of two innovative modules: the *Spatial Compact Module* (*SCM*) and the *Temporal Compact Module* (*TCM*). For the RGB-X dual input streams, SCM facilitates essential interaction between the two modalities to generate a compact spatial feature. Additionally, SCM introduces a small set of learnable queries to preserve modality-specific information. This compact visual feature, along with the queries, is then fed into a one-stream backbone network (Ye et al., 2022; Zhang et al., 2022b) to enable comprehensive intra- and inter-modality modeling. Next, TCM constructs the refined target distribution heatmap in the search image, selecting key target cues to model compact temporal features at each time step. Dense temporal features are then obtained by aggregating over multiple steps, serving as a supplement to sparse dynamic templates. Thanks to this compact spatiotemporal feature representation, CSTrack achieves new SOTA results on RGB-D/T/E tasks.

Our contributions are as follows:

- To address the limitations of RGB-X dual dispersed feature spaces, we propose a simple yet effective tracker, CSTrack, by modeling and utilizing novel compact multimodal spatiotemporal features.

- We design the innovative Spatial Compact Module and Temporal Compact Module, which leverage the compact spatiotemporal features for comprehensive intra- and inter-modality spatial modeling and dense temporal modeling.

- Through extensive experiments, we demonstrate the effectiveness of our proposed spatiotemporal compact approach. Furthermore, CSTrack achieves new SOTA performance on mainstream RGB-D/T/E benchmarks.

## 2. Related Works

### 2.1. Spatial Modeling in RGB-X Tracking

To leverage the complementary advantages of RGB and X modalities, existing RGB-X trackers typically use two parallel branches for spatial modeling(Zhang et al., 2024b). These trackers include intra-modality modeling between the search images and template cues of each modality, as well as inter-modality modeling through feature interaction between modalities (Wang et al., 2024b). Specifically, the RGB branch usually adopts the backbone of a pretrained RGB tracker (Ye et al., 2022), while the X branch follows different implementation strategies. Inspired by prompt learning (Lester et al., 2021), ViPT (Zhu et al., 2023a) and OneTracker (Hong et al., 2024) design a handcrafted X branch that treats the X modality as prompts, progressively embedding them into the RGB branch to facilitate inter-modality interaction. Additionally, SDSTrack (Hou et al., 2024) and BAT (Cao et al., 2024) adopt a symmetrical architecture, where the X branch is also derived from the pretrained RGB tracker and introduces adapter modules to enable knowledge transfer and inter-modality modeling. Managing the RGB-X dual dispersed feature spaces simultaneously introduces certain limitations, prompting us to propose a novel approach that integrates the RGB-X dual input streams into a compact feature space, enabling effective intra- and inter-modality spatial modeling.

### 2.2. Temporal Modeling in RGB-X Tracking

Due to the dynamic nature of the tracking target, the initially provided static template often fails to offer continuous guidance, especially when the target undergoes significant appearance changes (Feng et al., 2024; Zhang et al., 2024a). However, most existing RGB-X trackers still overly rely on the initial template, with only a few recent works incorporating temporal modeling to capture the target's evolving appearance cues. Notably, STMT (Sun et al., 2024) and TATrack (Wang et al., 2024a) introduce dynamic multimodal templates. However, since these templates encode information from a single time step, such sparse temporal modeling struggles to maintain robust tracking in challenging scenarios (Hu et al., 2024; 2023b). We argue that the

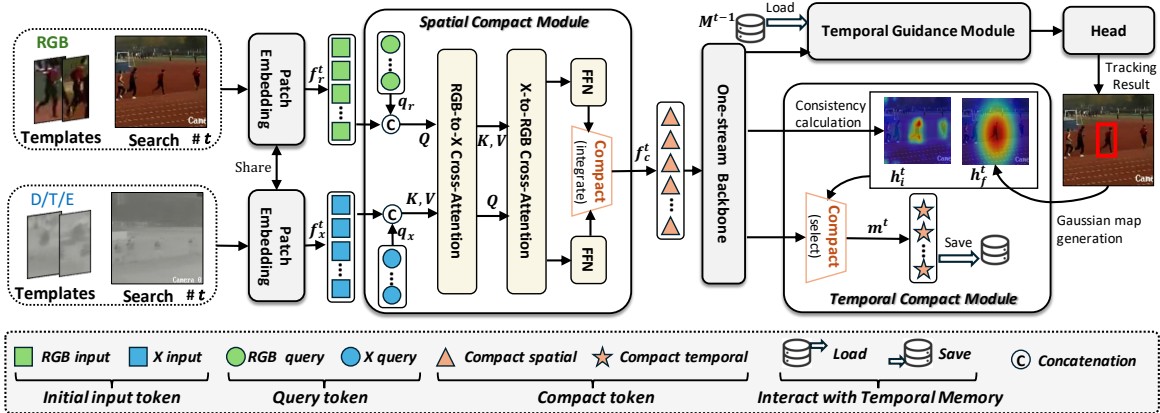

Figure 1: **Framework of our proposed CSTrack.** Given the RGB and X (*e.g.*, thermal data) input streams at time $t$ ($t \geq 1$), the shared Patch Embedding initially transforms them into token sequences. Then, the Spatial Compact Module integrates them into a compact feature space, which is subsequently fed into a One-stream Backbone for comprehensive spatial modeling. Next, the Temporal Guidance Module uses the previously stored temporal features (up to $t-1$) for tracking guidance, after which the Head generates the final tracking results. Subsequently, the Temporal Compact Module constructs compact temporal features for the current time step, which are stored for tracking guidance at the next time step $t+1$.

computational overhead of maintaining RGB-X dual dispersed feature spaces is a key factor limiting dense temporal modeling. For instance, in TATrack's online branch (Wang et al., 2024a), each additional time step requires spatiotemporal modeling in both the RGB and X branches, leading to a multiplicative increase in computational costs. What sets us apart is that our compact multimodal features exist in a single branch, facilitating efficient dense temporal modeling. Specifically, we devise a parameter-free temporal representation and a lightweight spatiotemporal fusion module, substantially enhancing the tracker's robustness in complex scenarios.

## 3. Methods

The framework of CSTrack is shown in Fig. 1. Given the RGB and X modality search images and template cues (initial and dynamic template images) at time $t$ ($t \geq 1$), the shared *Path Embedding* first organizes them into token sequences. The *Spatial Compact Module (SCM)* then generates a compact spatial feature by essentially interacting the two modalities, incorporating a few queries to preserve modality-specific information. This feature and the queries are fed into a *One-stream Backbone* for comprehensive intra- and inter-modality spatial modeling. Next, the *Temporal Guidance Module* uses the dense temporal features stored in temporal memory (up to $t-1$) to guide the tracking process, which is then passed to the *Head* to generate the tracking results. Finally, the *Temporal Compact Module (TCM)* selects key tokens from the search features to compactly represent temporal features, achieved through the target distribution maps derived from the intermediate and final results.

### 3.1. Spatial Compact Modeling

#### 3.1.1. SHARED PATCH EMBEDDING

At time step $t$, the RGB-X dual streams fed into the model consist of the following images: the search images $S_r^t, S_x^t \in \mathbb{R}^{3 \times H_s \times W_s}$, the initial template images $Z_r^0, Z_x^0 \in \mathbb{R}^{3 \times H_z \times W_z}$, and the dynamic template images $Z_r^t, Z_x^t \in \mathbb{R}^{3 \times H_z \times W_z}$. In line with the current ViT-based tracking paradigm (Ye et al., 2022), we first partition each image into patches and then convert them into token sequences (Dosovitskiy et al., 2020). We employ the patch embedding method from HiViT (Zhang et al., 2022b), which, as a variant of the ViT model, better preserves spatial information through gradual downsampling, making it widely adopted in recent trackers (Shi et al., 2024; Xie et al., 2024).

Compared to previous works that design separate patch embeddings for each modality (Zhu et al., 2023a; Hong et al., 2024), we adopt a shared patch embedding module across modalities. The insight behind this is that both RGB and X images are essentially visual signals (Yang et al., 2022), and joint learning of RGB, depth, thermal, and event data can enhance the model's multimodal perception capability (Zhang et al., 2023). After processing, we obtain the search features $s_r^t, s_x^t \in \mathbb{R}^{N_s \times D}$, the initial template features $z_r^0, z_x^0 \in \mathbb{R}^{N_z \times D}$, and the dynamic template features $z_r^t, z_x^t \in \mathbb{R}^{N_z \times D}$. We concatenate the visual information of each modality to obtain $f_r^t$ and $f_x^t \in \mathbb{R}^{N_{zs} \times D}$, with $N_{zs} = 2N_z + N_x$:

$$f_r^t = [z_r^0; z_r^t; s_r^t], f_x^t = [z_x^0; z_x^t; s_x^t]. \quad (1)$$

where [;] indicates concatenation along the first dimension.

### 3.1.2. SPATIAL COMPACT MODULE

The $f_r^t$ and $f_x^t$ represent two modality-specific and dispersed feature spaces, each containing long token sequences. Existing trackers typically adopt two parallel branches to handle these dispersed feature spaces (Hong et al., 2024; Hou et al., 2024), leading to complex model structures and high computational costs. To address this limitation, our proposed SCM integrates $f_r^t$ and $f_x^t$ into a compact spatial feature, enabling simplified and effective spatial modeling.

Specifically, we first introduce two small sets of learnable queries, $q_r, q_x \in \mathbb{R}^{N_q \times D}$, to preserve modality-specific information during the spatial compacting process. These queries are concatenated with the spatial features and participate in modality interaction via bidirectional cross-attention (Liu et al., 2025). Then, the features of each modality are further integrated through an feed-forward neural network.

$$[q'_r; f'^t_r] = Norm([q_r; f^t_r] + \Phi_{CA}([q_r; f^t_r], [q_x; f^t_x])), \tag{2}$$

$$[q'_x; f'^t_x] = Norm([q_x; f^t_x] + \Phi_{CA}([q_x; f^t_x], [q'_r; f'^t_r])), \tag{3}$$

$$[q''_r; f''^t_r] = Norm([q'_r; f'^t_r] + FFN([q'_r; f'^t_r])), \tag{4}$$

$$[q''_x; f''^t_x] = Norm([q'_x; f'^t_x] + FFN([q'_x; f'^t_x])). \tag{5}$$

Here, $\Phi_{CA}(\cdot, \cdot)$ represents the cross-attention operation where the first element serves as $Q$ and the second element serves to obtain $K$ and $V$. $Norm$ refers to the normalization operation, and $FFN$ denotes the feed-forward fully connected network (Vaswani et al., 2017).

Although the RGB modality contains rich detailed information and the X modality focuses more on edge and contour information (Wu et al., 2024), they are well-aligned in spatial semantics (Yang et al., 2022). As a result, the $f''^t_r$ and $f''^t_x$ remain spatially aligned. We obtain the preliminary compact spatial feature $f^t_{c0}$ by simply adding them together:

$$f^t_{c0} = f''^t_r + f''^t_x. \tag{6}$$

Additionally, the query tokens involved in the entire feature interaction process implicitly preserve the characteristics of each modality. We incorporate them to construct the final compact spatial feature $f^t_c \in \mathbb{R}^{(2N_q + N_{zs}) \times D}$:

$$f^t_c = [q''_r; q''_x; f^t_{c0}]. \tag{7}$$

### 3.1.3. ONE-STREAM BACKBONE

After the above processing, the length of the visual token sequence handled by the model is reduced from $2N_{zs}$ to $2N_q + N_{zs}$, where $N_q$ is much smaller than $N_{zs}$, meaning that the size of $f^t_c$ is comparable to that of a single modality. This allows us to use a one-stream Transformer-based backbone (Ye et al., 2022; Shi et al., 2024; Xie et al., 2024) to

further integrate the features of $f^t_c$.

$$f'^t_c = \Phi_{Backbone}(f^t_c). \tag{8}$$

Since $f^t_c$ contains information from both modalities, and the Transformer network excels in token interaction (Vaswani et al., 2017), this process effectively enables comprehensive intra- and inter-modality spatial modeling.

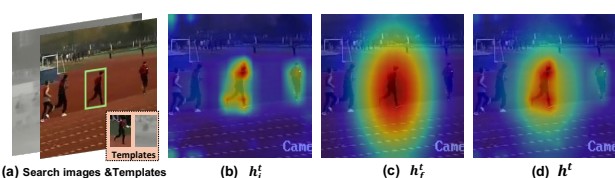

(a) Search images &Templates    (b) $h^t_i$    (c) $h^t_f$    (d) $h^t$

Figure 2: Illustration of different target heatmaps (using an RGB-T sample as an example). (a) Search images and templates, with the green bounding box indicating the target to be tracked. (b-d) Target distribution heatmaps derived from intermediate results, final results, and their combination, *i.e.*, $h^t_i$, $h^t_f$, and $h^t$ (reshaped into 2D images for visualization).

## 3.2. Temporal Compact Modeling

### 3.2.1. TEMPORAL COMPACT MODULE

In addition to the sparse temporal information provided by the multimodal dynamic template (Sun et al., 2024; Wang et al., 2024a), TCM constructs the temporal feature $m^t$ for the current time step and stores it in the temporal memory. By aggregating $m^t$ from multiple time steps, we obtain the dense temporal feature $M^t$ up to time step $t$, which will guide the tracking process for the next time step $t + 1$. The appearance information of the target in the search image reflects its dynamic changes (Ling et al., 2025), providing important guidance cues for the tracker (Zhou et al., 2024), and thus is regarded as the primary modeling object for $m^t$. To this end, TCM designs a parameter-free method for constructing the target distribution heatmap, which precisely reflects the target's position distribution in the search image. Then, we extract $N_m$ key target tokens from the integrated spatial feature $f'^t_c$ to obtain the compact temporal feature $m^t \in \mathbb{R}^{N_m \times D}$.

The target distribution heatmap consists of two parts, which are constructed based on the intermediate and final results, respectively. For the former, since $f'^t_c$ undergoes comprehensive feature integration through the backbone, the relationships between the search and template features are well-modeled. Thus, the consistency of the search tokens with respect to the template features can be computed to generate a target heatmap $h^t_i$ based on the intermediate results. In this process, we first index the search feature $s^t_c \in \mathbb{R}^{N_s \times D}$ and template feature $z^t_c \in \mathbb{R}^{2N_z \times D}$ from $f'^t_c$ based on the feature concatenation order in Eq. 1 and Eq. 7. Then, we per-

form correlation-like operations on $s_c^t$ to refine the features (Feng et al., 2025):

$$s_c'^t = s_c^t \cdot (s_c^t)^T \cdot s_c^t. \tag{9}$$

Next, a dot product operation between $s_c'^t$ and $z_c^t$ is performed to obtain the target heatmap $h_i^t \in \mathbb{R}^{N_s \times 1}$:

$$h_i^t = (s_c'^t \cdot (z_c^t)^T).mean(dim = 1). \tag{10}$$

As shown in Fig. 2, $h_i^t$ finely depicts the target's position distribution, which is crucial for selecting target tokens. However, since $h_i^t$ is built through similarity matching with the target appearance in $z_c^t$, it may also focus on objects similar to the target, potentially causing interference to the tracker. To address this, we introduce another heatmap, $h_f^t$, based on the final result. It is constructed using the predicted bounding box $b^t = (x_c, y_c, w, h)$, obtained from the subsequent Head module (described in Sec. 3.3). Here, $(x_c, y_c), (w, h)$ represent the target's position and size in the search image, respectively. Based on the target-only position information contained in $b^t$, $h_{f0}^t \in \mathbb{R}^{W_s \times H_s}$ is constructed by converting it into a 2D Gaussian distribution, as expressed by:

$$h_{f0}^t(x, y) \propto \exp\left( -\frac{1}{2} \left[ \left( \frac{x - x_c}{w/3} \right)^2 + \left( \frac{y - y_c}{h/3} \right)^2 \right] \right). \tag{11}$$

where $(x_c, y_c)$ is considered the mean, and $(w, h)$ represents three times the standard deviation. By downsampling $h_{f0}^t$ to match the 2D scale of $s_c^t$ and flattening it, we obtain $h_f^t \in \mathbb{R}^{N_s \times 1}$. Compared to $h_i^t$, as shown in Fig. 2, $h_f^t$ mainly carries the target's position distribution information, but it is not as finely detailed as $h_i^t$. To leverage the strengths of both, we compute the weighted sum of the two to obtain the final target distribution map $h^t \in \mathbb{R}^{N_s \times 1}$:

$$h^t = 0.5 \times Norm(h_i^t) + 0.5 \times Norm(h_f^t). \tag{12}$$

The $h^t$ reflects the likelihood of each token in $s_c^t$ belonging to the target. Based on this, we select the top-$N_m$ tokens from $s_c^t$ as the compact temporal feature $m^t$ for this time step.

Subsequently, $m^t$ is stored in the temporal memory to form the multi-step dense temporal feature $M^t = \{m^i\}_{i=1}^L$. $M^t$ is a buffer of length $L$, initialized with $m^1$ at the starting time, and updated by a sliding window mechanism (Xie et al., 2024; Cai et al., 2024) to store the $L$ most recent $m^t$ (See Sec. C.3 for details).

### 3.2.2. TEMPORAL GUIDANCE MODULE

This module is primarily designed to utilize the temporal feature $M^{t-1}$ stored prior to the current time step $t$ to guide

tracking. Specifically, we first concatenate the individual temporal feature units in $M^{t-1}$ to obtain $M'^{t-1}$.

Then, we employ a vanilla transformer-based decoder (Vaswani et al., 2017) to facilitate the interaction between the search feature and $M'^{t-1}$, embedding the temporal variation cues of the target appearance into the search feature:

$$s_c'^t = Norm(s_c^t + \Phi_{CA}(s_c^t, M'^{t-1})), \tag{13}$$

$$s_{cm}^t = Norm(s_c'^t + FFN(s_c'^t)). \tag{14}$$

### 3.3. Head and Loss

Based on the search feature $s_{cm}^t$, which integrates the template and temporal cues, we employ a classic CNN-based prediction head (Ye et al., 2022; Xie et al., 2024) to obtain the final bounding box $b^t$. To supervise the prediction of the bounding box, we employ the widely used focal loss $L_{cls}$ (Law & Deng, 2018), $L_1$ loss, and the generalized IoU loss $L_{iou}$ (Rezatofighi et al., 2019). The overall loss function is formulated as follows:

$$L_{all} = L_{cls} + \lambda_{iou} L_{iou} + \lambda_{L_1} L_1, \tag{15}$$

where $\lambda_{iou} = 2$ and $\lambda_{L_1} = 5$ are the specific regularization parameters.

## 4. Experiments

### 4.1. Implementation Details

For the implementation of CSTrack, we adopt HiViT-base (Zhang et al., 2022b) as the modality-shared patch embedding and backbone. The CSTrack is initialized with Fast-iTPN (Tian et al., 2024) pre-trained weights, and the token dimension $D$ is set to 512. which are initialized with the Fast-iTPN pre-trained parameters and the token dimension D set to 512. In the SCM, the length of modality-specific queries $N_q$ is set to 4. In the TCM, each temporal feature is represented by 16 tokens (i.e., $N_m = 16$), with the temporal length $L$ set to 4 by default. The sizes of template patches and search images are $128 \times 128$ and $256 \times 256$, respectively. Our tracker is trained on a server equipped with four A5000 GPUs and tested on an RTX-3090 GPU. CSTrack consists of 75M parameters and achieves a tracking speed of 33 FPS.

Inspired by recent unified models (Chen et al., 2022a; Yan et al., 2023), we adopt a joint training approach to develop a tracker capable of handling RGB-D/T/E tasks simultaneously. First, our training dataset includes common RGB-X datasets such as DepthTrack (Yan et al., 2021b), LaSHeR (Li et al., 2021) and VisEvent (Wang et al., 2023). Furthermore, considering that the scale of these datasets is insufficient to support joint training, we also incorporate widely used RGB tracking datasets, namely LaSOT (Fan et al., 2019), GOT-10K (Huang et al., 2019), COCO (Lin et al., 2014),

Table 1: Comparison on RGB-Depth datasets. The top two results are highlighted in red and blue, respectively.

| Method | DepthTrack | | | VOT-RGBD22 | | |
|---|---|---|---|---|---|---|
| | F-score | Re | PR | EAO | Acc | Rob |
| DAL (Qian et al., 2021) | 42.9 | 36.9 | 51.2 | - | - | - |
| LTMU-B (Dai et al., 2020) | 46.0 | 41.7 | 51.2 | - | - | - |
| ATCAIS (Kristan et al., 2020) | 47.6 | 45.5 | 50.0 | 55.9 | 76.1 | 73.9 |
| DRefine (Kristan et al., 2021) | - | - | - | 59.2 | 77.5 | 76.0 |
| KeepTrack (Mayer et al., 2021) | - | - | - | 60.6 | 75.3 | 79.7 |
| DMTrack (Kristan et al., 2022) | - | - | - | 65.8 | 75.8 | 85.1 |
| DeT (Yan et al., 2021b) | 52.9 | 54.3 | 56.0 | 65.7 | 76.0 | 84.5 |
| SPT (Zhu et al., 2023b) | 53.8 | 54.9 | 52.7 | 65.1 | 79.8 | 85.1 |
| ProTrack (Yang et al., 2022) | 57.8 | 57.3 | 58.3 | 65.1 | 80.1 | 82.0 |
| ViPT (Zhu et al., 2023a) | 59.6 | 59.4 | 59.2 | 72.1 | 81.8 | 86.7 |
| UnTrack (Wu et al., 2024) | 61.0 | 60.6 | 61.1 | 72.1 | 82.0 | 86.9 |
| OneTracker (Hong et al., 2024) | 60.9 | 60.4 | 60.7 | 72.7 | 81.9 | 87.2 |
| SDSTrack (Hou et al., 2024) | 61.9 | 60.1 | 61.4 | 72.8 | 81.2 | 88.3 |
| **CSTrack** | 65.8 | 66.4 | 65.2 | 77.4 | 83.3 | 92.9 |

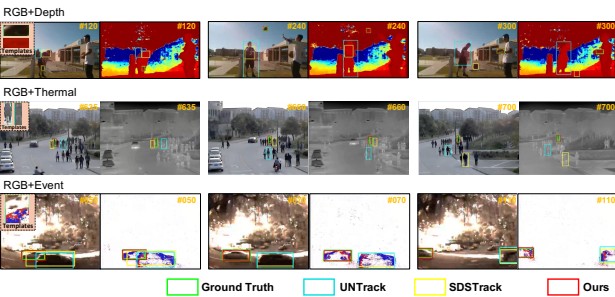

RGB+Depth

RGB+Thermal

RGB+Event

Ground Truth    UNTrack    SDSTrack    Ours

Figure 3: Qualitative comparison results of our tracker with other two trackers (*i.e.*, UNTrack and SDSTrack) on three challenging cases. Better viewed in color with zoom-in.

TrackingNet (Muller et al., 2018), VastTrack (Peng et al., 2024), and TNL2k (Wang et al., 2021b), into our training set. For the RGB datasets, we input RGB images into both the RGB and X interfaces of the model to achieve consistency in input formats with RGB-X datasets. We further analyze datasets from different modalities. Results show that the model is capable of effectively distinguishing between different modalities (see Sec. B for more details).

Our training process consists of two stages. In the first stage, we train the model for 150 epochs without incorporating TCM. Each epoch contains 10,000 samples, and each sample consists of a single search image. In the second stage, we introduce TCM based on the model trained in the first stage, which only involves spatial compactness modeling. During this stage, the backbone of the model is frozen, and we train the remaining parameters for 50 epochs. Each epoch includes 3,000 samples, where each sample comprises six search images. We employ the AdamW optimizer to optimize the network parameters. During the inference phase, the dynamic template update follows the strategy proposed in TATrack (Wang et al., 2024a). For benchmarks of different modalities, we adopt different thresholds for dynamic template updating (see Sec. B for more details).

Table 2: Comparison on RGB-Thermal datasets.

| Method | LasHeR | | RGBT234 | |
|---|---|---|---|---|
| | SR | PR | MSR | MPR |
| SGT (Li et al., 2017) | 25.1 | 36.5 | 47.2 | 72.0 |
| DAFNet (Gao et al., 2019) | - | - | 54.4 | 79.6 |
| FANet (Zhu et al., 2020) | 30.9 | 44.1 | 55.3 | 78.7 |
| MaCNet (Zhang et al., 2020) | - | - | 55.4 | 79.0 |
| HMFT (Zhang et al., 2022a) | 31.3 | 43.6 | - | - |
| CAT (Li et al., 2020) | 31.4 | 45.0 | 56.1 | 80.4 |
| DAPNet (Zhu et al., 2019) | 31.4 | 43.1 | - | - |
| JMMAC (Zhang et al., 2021) | - | - | 57.3 | 79.0 |
| CMPP (Wang et al., 2020) | - | - | 57.5 | 82.3 |
| APFNet (Xiao et al., 2022) | 36.2 | 50.0 | 57.9 | 82.7 |
| OSTrack (Ye et al., 2022) | 41.2 | 51.5 | 54.9 | 72.9 |
| ProTrack (Yang et al., 2022) | 42.0 | 53.8 | 59.9 | 79.5 |
| ViPT (Zhu et al., 2023a) | 52.5 | 65.1 | 61.7 | 83.5 |
| TBSI (Hui et al., 2023) | 55.6 | 69.2 | 63.7 | 87.1 |
| BAT (Cao et al., 2024) | 53.1 | 66.5 | 62.5 | 84.8 |
| UnTrack (Wu et al., 2024) | 51.3 | 64.6 | 62.5 | 84.2 |
| SDSTrack (Hou et al., 2024) | 53.1 | 66.5 | 62.5 | 84.8 |
| OneTracker (Hong et al., 2024) | 53.8 | 67.2 | 64.2 | 85.7 |
| IPL (Lu et al., 2024) | 55.3 | 69.4 | 65.7 | 88.3 |
| **CSTrack** | 60.8 | 75.6 | 70.9 | 94.0 |

## 4.2. Comparison with State-of-the-arts

### 4.2.1. QUANTITATIVE COMPARISON

We evaluate CSTrack on mainstream benchmarks and provide a thorough comparison with SOTA models. Specifically, Tab. 1 presents the evaluation of performance in the RGB-Depth task using DepthTrack (Yan et al., 2021b) and VOT-RGBD2022 (Kristan et al., 2022); Tab. 2 evaluates the RGB-Thermal task using LasHeR (Li et al., 2021) and RGBT234 (Li et al., 2019); and Tab. 3 illustrates the model's performance in the RGB-Event task, evaluated with VisEvent (Wang et al., 2023). For more information on these benchmarks, refer to Sec. A.

As shown in Tables 1, 2, and 3, CSTrack achieves new SOTA across all benchmarks, significantly outperforming previous best models. Taking precision rate (PR) as an example, CSTrack surpasses SDSTrack (Hou et al., 2024) by 3.8% in DepthTrack, IPL (Lu et al., 2024) by 6.2% in LasHeR, and OneTracker (Hong et al., 2024) by 5.7% in VisEvent. These outstanding results demonstrate the effectiveness and generalization capability of our approach. Compared to existing trackers that separately handle the RGB and X modality feature spaces, CSTrack's distinct advantage lies in modeling and utilizing a compact spatiotemporal feature. These results strongly validate the effectiveness of our approach.

Additionally, the recent PIL (Lu et al., 2024) introduces the challenge of modality missing to address more realistic practical scenarios. Specifically, it proposes the LasHeR-Miss and RGBT234-Miss datasets for the RGB-Thermal task. To evaluate CSTrack, we adopted a simple method by cloning the available modality image onto the missing modality image. As shown in Tab. 4, although we did not design a dedi-

Table 3: Comparison on the RGB-Event dataset.

| Method | VisEvent | |
| --- | --- | --- |
| | SR | PR |
| SiamMask_E (Wang et al., 2019) | 36.9 | 56.2 |
| SiamBAN_E (Chen et al., 2022b) | 40.5 | 59.1 |
| ATOM_E (Danelljan et al., 2019) | 41.2 | 60.8 |
| VITAL_E (Song et al., 2018) | 41.5 | 64.9 |
| SiamCar_E (Guo et al., 2020) | 42.0 | 59.9 |
| MDNet_E (Nam & Han, 2016) | 42.6 | 66.1 |
| STARK_E (Yan et al., 2021a) | 44.6 | 61.2 |
| PrDiMP_E (Danelljan et al., 2020) | 45.3 | 64.4 |
| LTMU_E (Dai et al., 2020) | 45.9 | 65.5 |
| TransT_E (Chen et al., 2021) | 47.4 | 65.0 |
| SiamRCNN_E (Voigtlaender et al., 2020) | 49.9 | 65.9 |
| OSTrack (Ye et al., 2022) | 53.4 | 69.5 |
| ProTrack (Yang et al., 2022) | 47.1 | 63.2 |
| ViPT (Zhu et al., 2023a) | 59.2 | 75.8 |
| UnTrack (Wu et al., 2024) | 58.9 | 75.5 |
| OneTracker (Hong et al., 2024) | 60.8 | 76.7 |
| SDSTrack (Hou et al., 2024) | 59.7 | 76.7 |
| **CSTrack** | 65.2 | 82.4 |

Table 4: Comparison on RGB-Thermal modality-missing datasets.

| Method | LasHeR-Miss | | RGBT234-Miss | |
| --- | --- | --- | --- | --- |
| | SR | PR | MSR | MPR |
| CAT (Li et al., 2020) | 20.6 | 28.5 | 35.6 | 52.1 |
| APFNet (Xiao et al., 2022) | 29.0 | 38.2 | 44.7 | 65.3 |
| ViPT (Zhu et al., 2023a) | 36.9 | 44.0 | 44.8 | 61.2 |
| TBSI (Hui et al., 2023) | 47.0 | 59.0 | 44.8 | 61.2 |
| BAT (Cao et al., 2024) | 44.0 | 54.5 | 52.8 | 73.4 |
| SDSTrack (Hou et al., 2024) | 44.4 | 54.8 | 52.5 | 72.4 |
| IPL (Lu et al., 2024) | 49.4 | 61.7 | 59.4 | 82.0 |
| **CSTrack** | 50.3 | 62.2 | 68.0 | 89.8 |

cated module for modality missing as PIL did, our method still demonstrates a significant performance advantage. For instance, on the RGBT234-Miss, CSTrack surpasses IPL by 8.6% in mean success rate (MSR), showcasing the robustness of CSTrack to missing input information (Chen et al., 2024; Feng et al., 2023).

### 4.2.2. QUALITATIVE COMPARISON

As shown in Fig. 3, we present the tracking results of CSTrack and two existing SOTA trackers (*i.e.*, SDSTrack (Hou et al., 2024), UnTrack (Wu et al., 2024)) on the RGB-D/T/E challenging sequences. More cases can be found in Sec. D. Clearly, CSTrack demonstrates superior robustness and effectiveness.

### 4.3. Ablation Study

To investigate the properties of the compact spatiotemporal modeling approach in CSTrack, we conduct comprehensive ablation studies on the DepthTrack (Yan et al., 2021b), RGBT234 (Li et al., 2019) and VisEvent (Wang et al., 2023)

benchmarks. For implementation details of each ablation setting, please refer to Sec. C.

### 4.3.1. STUDY ON OUR SPATIAL COMPACT METHOD

As described in Sec. 3.1, the primary motivation for our compact spatial modeling is that existing methods handling RGB-X dual feature spaces introduce complex model structures and computation processes, limiting the effectiveness of multimodal spatial modeling. To validate this, we conduct ablation studies on various spatial modeling architectures. First, for the RGB-X dual-branch architecture, we implement two representative asymmetric (Zhu et al., 2023a; Hong et al., 2024) and symmetric (Hou et al., 2024; Cao et al., 2024) models (See Sec. C.1 for details). Next, we remove the TCM from CSTrack, treating it as the baseline model for compact spatial modeling. To ensure fairness, we use the same backbone network and training strategy.

Tab. 5 shows the performance and computational efficiency of different frameworks. Compared to #1 and #2, the symmetric dual-branch architecture achieves better performance, consistent with the findings in SDSTrack (Hou et al., 2024), but at the cost of higher parameters and computation. In contrast, our method (#3) achieves superior performance while significantly reducing computational overhead, demonstrating the effectiveness of our compact spatial modeling.

Additionally, we perform the ablation studies on different core components of SCM. As shown in Tab. 6, #1 represents the baseline model for our compact spatial modeling. #2 indicates the performance after removing modality-specific queries (*i.e.*, $q_r$, $q_x$), where a performance degradation is observed across all three benchmarks. We attribute this to these queries implicitly capturing key information from each modality, enabling the effective construction of compact spatial features. #3 indicates that we replace the inter-modality cross-attention (in Eq. 2 and Eq. 3) with self-attention within each modality. #4 represents the setting where we do not use the shared patch embedding but design separate embeddings for RGB and X. Compared to RGBT234 and VisEvent, DepthTrack is more challenging, as evidenced by its lower absolute precision. We hypothesize that modality interaction before compression and learning a unified representation through shared embeddings contribute to better performance in difficult tracking scenarios.

Finally, Fig. 4 provides a perspective for visual analysis. We analyze two representative cases. RGB(X) advantage indicates that the tracker primarily relies on RGB(X) modality, as target appearance information in the other modality, X(RGB), is significantly degraded due to environmental interference (*e.g.*, overlapping similar objects or lighting disturbance). Correspondingly, the model achieves accurate tracking with only RGB(X) input but fails with only X(RGB) input. As shown in Fig. 4 (d), the model delivers

Table 5: Comparison of Different RGB-X Spatial Modeling Frameworks.

| # | Frameworks | RGBT234 | | DepthTrack | | | VisEvent | | Params | FLOPs | Speed |
|---|---|---|---|---|---|---|---|---|---|---|---|
| | | MSR | MPR | Re | PR | SR | PR | F-score | | | |
| 1 | Dual-Branch (Asymmetrical) | 65.9 | 92.0 | 62.7 | 63.2 | 62.2 | 62.8 | 79.1 | 82M | 38G | 24 FPS |
| 2 | Dual-Branch (Symmetrical) | 68.2 | 92.4 | 63.6 | 64.1 | 63.1 | 62.9 | 79.6 | 136M | 59G | 18 FPS |
| 3 | One Compact Branch (ours) | 69.6 | 93.0 | 65.1 | 65.3 | 64.9 | 64.1 | 81.0 | 73M | 36G | 35 FPS |

Table 6: Ablation Study on Spatial Compact Module.

| # | Setting | RGBT234 | | DepthTrack | | | VisEvent | |
|---|---|---|---|---|---|---|---|---|
| | | MSR | MPR | F-score | Re | PR | SR | PR |
| 1 | Baseline | 69.6 | 93.0 | 65.1 | 65.3 | 64.9 | 64.1 | 81.0 |
| 2 | *w/o* queries | 67.3 | 92.0 | 64.1 | 64.5 | 63.7 | 63.7 | 80.3 |
| 3 | *w/o* cross-attention | 69.5 | 92.8 | 60.6 | 61.0 | 60.2 | 64.1 | 81.0 |
| 4 | unshare embedding | 69.2 | 93.0 | 61.1 | 61.3 | 60.9 | 64.1 | 80.8 |

Table 8: Ablation Study on Training Strategies.

| # | Setting | RGBT234 | | DepthTrack | | | VisEvent | |
|---|---|---|---|---|---|---|---|---|
| | | MSR | MPR | F-score | Re | PR | SR | PR |
| 1 | Baseline | 69.6 | 93.0 | 65.1 | 65.3 | 64.9 | 64.1 | 81.0 |
| 2 | *w/o* RGB datasets | 61.9 | 85.9 | 60.9 | 61.2 | 60.6 | 61.8 | 79.3 |
| 3 | *w/o* joint train | 67.6 | 90.9 | 64.2 | 64.7 | 63.8 | 63.0 | 80.3 |

Table 7: Ablation Study on Temporal Compact Module.

| # | Setting | RGBT234 | | DepthTrack | | | VisEvent | |
|---|---|---|---|---|---|---|---|---|
| | | MSR | MPR | F-score | Re | PR | SR | PR |
| 1 | Baseline | 69.6 | 93.0 | 65.1 | 65.3 | 64.9 | 64.1 | 81.0 |
| 2 | *w* RoI-based | 69.0 | 92.8 | 64.4 | 64.9 | 63.9 | 64.3 | 81.1 |
| 3 | *w* Query-based | 69.4 | 92.2 | 63.3 | 63.7 | 62.8 | 64.1 | 80.9 |
| 4 | *w* $h_i^t$ | 70.1 | 93.6 | 64.8 | 65.0 | 64.6 | 64.9 | 82.1 |
| 5 | *w* $h_f^t$ | 70.2 | 93.8 | 64.2 | 64.4 | 64.0 | 64.8 | 82.0 |
| 6 | *w* $h^t$ ($h_i^t$ & $h_f^t$) | 70.9 | 94.0 | 65.8 | 66.4 | 65.2 | 65.2 | 82.4 |

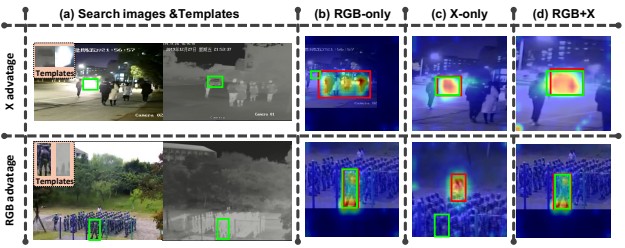

Figure 4: Tracking results of the model under different input settings in two categories of cases (using the RGB-T task as an example). (a) Search images and templates. (b-d) Tracking results with only RGB input, only X input, and both inputs. The heatmap regions are cropped by the tracker. The green and red bounding boxes represent the target to be tracked and the tracking result. Better viewed with zoom-in.

precise tracking results when RGB+X inputs are available. This indirectly demonstrates that, even though spatial information from both modalities is compacted together, critical information from each modality is retained, validating the effectiveness of our spatial compact modeling.

### 4.3.2. STUDY ON OUR TEMPORAL COMPACT METHOD

Our TCM focuses on selecting a small set of key target tokens from the search features as compact temporal features. To evaluate this, we conduct the ablation studies on different temporal compacting methods, as shown in Tab. 7. Inspired by existing works, we first explore two widely used methods: #2 involves using predicted bounding boxes for RoI processing in the search features (Wang et al., 2021a; Zhou et al., 2023), and #3 applies temporal query compression (Zheng et al., 2024; Xie et al., 2024). For detailed implementation, please refer to Sec. C.2. Compared to the baseline (#1), both methods lead to performance degradation. We hypothesize that this is due to the introduction of new feature variations for the search tokens, which places the constructed temporal features in a different feature space, thus failing to provide effective guidance for the tracker in the subsequent temporal guidance module. In contrast, our temporal features are selected directly from the search tokens without any feature variation. #4 and #5 represent the selection of search tokens using target heatmaps $h_i^t$ and $h_f^t$, both of which lead to performance improvements. As observed in Fig. 2, by combining the two heatmaps, we achieve a more optimal target distribution representation, which further enhances

model performance (#6).

As shown in Fig. 5, we analyze the impact of the temporal memory length $L$ on model performance. We observe that as the $L$ increases, the performance initially improves and then stabilizes. To balance performance and computational cost, we select 4 as the optimal temporal length.

### 4.3.3. OTHER ABLATION ANALYSES

To develop a tracker that simultaneously handles RGB-D/T/E tasks, we adopt a strategy of joint training on both RGB and X datasets, which significantly differs from existing approaches. As shown in Tab. 8, we conduct the ablation studies on this strategy. #2 indicates joint training using only the X datasets, resulting in a noticeable performance drop. The precision decreases by 7.1%, 4.3%, and 1.7% across the three benchmarks, highlighting the importance of large-scale RGB tracking datasets in joint training. #3 represents a non-joint training strategy, where a separate tracker is trained for each X modality. Although the number of task types is reduced, the loss of cross-modal knowledge sharing leads to a decrease in performance.

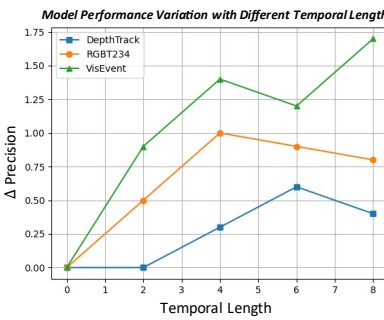

Figure 5: Model performance variation ($\Delta$ precision) with different temporal lengths.

## 5. Conclusions

To address the limitations of existing methods in handling RGB-X dual dispersed feature spaces, we propose a novel tracker named CSTrack, focusing on compact spatiotemporal feature modeling. For spatial modeling, we introduce the innovative Spatial Compact Module, which integrates the RGB-X dual input streams into a compact spatial feature, enabling comprehensive intra- and inter-modality spatial modeling. For temporal modeling, we design the efficient Temporal Compact Module to compactly represent temporal features, laying a solid foundation for dense temporal modeling. Through these combined efforts, CSTrack achieves outstanding performance, significantly outperforming existing methods on mainstream RGB-D/T/E benchmarks. Extensive experiments underscore the effectiveness of the compact spatiotemporal modeling method and provide a unique perspective for future RGB-X tracker design.

**Limitations and future work.** Our CSTrack effectively improves RGB-X tracking performance through a compact spatio-temporal modeling mechanism. Although this work covers auxiliary modalities in the form of visual data, namely depth, thermal, and event modalities, there is currently a lack of research on visual-language tracking (*i.e.*, RGB-L) that leverages text-based auxiliary modalities (Hu et al., 2023a; Li et al., 2024a;b;d;c). In the future, we plan to further investigate how to efficiently utilize textual modality cues to develop a tracker capable of accommodating all existing auxiliary modalities.

## Acknowledgements

This work is supported in part by the National Science and Technology Major Project (Grant No.2022ZD0116403), and the National Natural Science Foundation of China (Grant No.62176255).

## Impact Statement

This paper presents work whose goal is to advance the field of Machine Learning. There are many potential societal consequences of our work, none which we feel must be specifically highlighted here.

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

## A. More Details on the RGB-X Benchmarks

As discussed in Sec. 4.2.1, we perform a comprehensive evaluation of CSTrack on mainstream RGB-X benchmarks. Specifically, the model's performance on the RGB-Depth task is evaluated using DepthTrack (Yan et al., 2021b) and VOT-RGBD2022 (Kristan et al., 2022). For the RGB-Thermal task, we adopt LasHeR (Li et al., 2021) and RGBT234 (Li et al., 2019), while the RGB-Event task is assessed based on VisEvent (Wang et al., 2023). In this section, we provide an overview of these benchmarks and their respective evaluation metrics.

**DepthTrack.** DepthTrack (Yan et al., 2021b) is a comprehensive and long-term RGB-D tracking benchmark. It consists of 150 training sequences and 50 testing sequences, with 15 per-frame attributes. The evaluation metrics include precision rate (PR), recall (Re), and F-score.

**VOT-RGBD2022.** VOT-RGBD2022 (Kristan et al., 2022) is the latest benchmark in RGB-D tracking, consisting of 127 short-term RGB-D sequences designed to explore the role of depth in RGB-D tracking. This dataset employs an anchor-based short-term evaluation protocol (Kristan et al., 2020), which requires trackers to restart multiple times from different initialization points. The main performance metrics include Accuracy (Acc), Robustness (Rob), and Expected Average Overlap (EAO).

**LasHeR.** LasHeR (Li et al., 2021) is a large-scale, high-diversity benchmark for short-term RGB-T tracking, comprising 979 video pairs for training and 245 pairs for testing. The dataset evaluates tracker performance using precision rate (PR) and success rate (SR).

**RGBT234.** RGBT234 (Li et al., 2019) is a large-scale RGB-T tracking benchmark consisting of 234 pairs of visible and thermal infrared video sequences. The dataset employs mean success rate (MSR) and mean precision rate (MPR) for performance evaluation.

**VisEvent.** VisEvent (Wang et al., 2023) is the largest dataset for RGB-E tracking, consisting of 500 pairs of training videos and 320 pairs of testing videos. This dataset employs success rate (SR) and precision rate (PR) for performance evaluation.

## B. More Details on Model Implementation

As discussed in Sec. 4.1, we perform joint training using datasets from both RGB-only and RGB-X modalities. Here, we conduct statistical analyses on representative RGB and RGB-D/T/E datasets to obtain the mean values used for input normalization. These values are computed over the training videos of each dataset. For comparison, we also include the ImageNet dataset (Deng et al., 2009) as a reference representing natural image distributions. As shown in Tab. 9, for the RGB modality, LaSOT and DepthTrack—both collected in typical natural environments—exhibit mean values that are similar to those of ImageNet. In contrast, Lasher, which often focuses on dark scenes, and VisEvent, which predominantly contains yellow-tinted RGB images, show noticeable deviations from the natural image statistics. For the X modality, as illustrated in Fig. 3 of the paper, the image styles vary significantly across datasets, leading to substantial differences in the corresponding mean values.

Table 9: Statistical analysis of RGB and X modalities across different datasets.

| Dataset | RGB Mean | X Mean |
|---|---|---|
| ImageNet | 0.485, 0.456, 0.406 | – |
| LaSOT (RGB-only) | 0.456, 0.459, 0.426 | – |
| DepthTrack (RGB-D) | 0.417, 0.414, 0.393 | 0.574, 0.456, 0.240 |
| Lasher (RGB-T) | 0.500, 0.499, 0.471 | 0.372, 0.372, 0.368 |
| VisEvent (RGB-E) | 0.418, 0.375, 0.317 | 0.935, 0.904, 0.949 |

In addition, we further analyze the model's discriminative ability across modality datasets. Specifically, based on the features from the one-stream backbone (see Eq. 8 in the paper), we extract the tokens associated with learnable queries and construct a four-class classifier using two fully connected layers to identify the input modality type: RGB-only, RGB-D, RGB-T, and RGB-E. The existing model is frozen, and the classification head is trained for 2 epochs. The accuracy results shown

in Tab. 10 demonstrate that the model can effectively distinguish between different modality datasets, owing to the clear differences in their feature distributions.

Table 10: Classification accuracy (%) of modality type prediction across different datasets.

| Dataset | LaSOT (RGB-only) | DepthTrack (RGB-D) | Lasher (RGB-T) | VisEvent (RGB-E) |
|---|---|---|---|---|
| **Accuracy** | 100% | 98% | 100% | 100% |

During the inference phase, our dynamic template update follows the strategy proposed in TATrack (Wang et al., 2024a). For benchmarks of different modalities, we adopt modality-specific thresholds for dynamic template updating. Specifically, for RGB-T benchmarks, namely Lasher and RGBT-234, we set the update threshold to 0.45; for other datasets, the threshold is set to 0.7.

## C. Experimental Details of Ablation Studies

### C.1. Implementation of Different RGB-X Spatial Modeling Frameworks

As shown in Tab. 5 of Sec. 4.3.1, we compare our proposed spatial compact modeling method with existing dual-branch-based methods. For the latter, we adopt the existing approach and implement both symmetric (Hou et al., 2024; Cao et al., 2024) and asymmetric (Zhu et al., 2023a; Hong et al., 2024) models using the same backbone network and training strategy as CSTrack. Our compact spatial modeling method, through the proposed Spatial Compact Module, integrates the features of both RGB and X modalities into a compact feature space. This allows subsequent intra- and inter-modality spatial modeling to be performed using a single-branch network (*i.e.*, Our One-stream Backbone in Sec. 3.1.3). In contrast, the dual-branch meth-

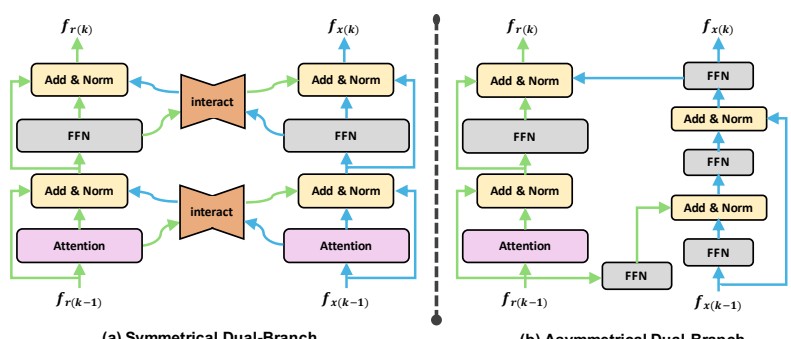

Figure 6: Processing workflows of symmetric and asymmetric dual-branch architectures, illustrated using the computational process of their respective $k$-th network layer as an example.

ods require separate handling of the RGB and X modality features, using two parallel branches for intra-modality spatial modeling and designing inter-modality feature interaction operations.

Specifically, the spatial features $f_r^t$ and $f_x^t$ of the two modalities (as described in Eq. 1, with the superscript $t$ omitted for simplicity) are obtained through a shared patch embedding and then input into the RGB and X branches, respectively. Below, we describe the feature processing methods at each layer in the dual-branch-based approaches.

**Dual-Branch (Symmetrical).**    This architecture employs two identical backbone networks as branches to process the two modalities. Consistent with CSTrack, we implement it using HiViT-base (Zhang et al., 2022b). Additionally, feature interaction between the two modalities is achieved by incorporating relevant modules at each layer, utilizing the widely adopted implementation from BAT (Cao et al., 2024). Taking the modeling process at the $k$-th layer of both branches as an

example (see Fig. 6 (a)), the specific computation is as follows:

$$f'_{r(k-1)} = \Phi^{RGB}_{CA(k)}(f_{r(k-1)}, f_{r(k-1)}), \tag{16}$$

$$f'_{r(k-1)} = Norm(f_{r(k-1)} + f'_{r(k-1)} + \Psi_{CA(k)}(f_{x(k-1)})), \tag{17}$$

$$f'_{x(k-1)} = \Phi^{X}_{CA(k)}(f_{x(k-1)}, f_{x(k-1)}), \tag{18}$$

$$f'_{x(k-1)} = Norm(f_{x(k-1)} + f'_{x(k-1)} + \Psi_{CA(k)}(f_{r(k-1)})), \tag{19}$$

$$f''_{r(k-1)} = FFN^{RGB}_{(k)}(f'_{r(k-1)}), \tag{20}$$

$$f_{r(k)} = Norm(f'_{r(k-1)} + f''_{r(k-1)} + \Psi_{FFN(k)}(f'_{x(k-1)})), \tag{21}$$

$$f''_{x(k-1)} = FFN^{X}_{(k)}(f'_{x(k-1)}), \tag{22}$$

$$f_{x(k)} = Norm(f'_{x(k-1)} + f''_{x(k-1)} + \Psi_{FFN(k)}(f'_{r(k-1)})). \tag{23}$$

In the feature modeling process at the $k$-th layer, the input consists of the output from the previous layer $(k-1)$, namely $f_{r(k-1)}$ and $f_{x(k-1)}$. For the RGB branch, the $k$-th layer includes a transformer-based attention network $\Phi^{RGB}_{CA(k)}$ and a feed-forward network $FFN^{RGB}_{(k)}$. Similarly, the symmetric X branch comprises $\Phi^{X}_{CA(k)}$ and $FFN^{X}_{(k)}$. To enable cross-modal interaction, we introduce two interaction modules, $\Psi_{CA(k)}$ and $\Psi_{FFN(k)}$, following the attention and feed-forward operations. Both $\Psi_{CA(k)}$ and $\Psi_{FFN(k)}$ are constructed using 3-layer fully connected networks. Finally, we obtain the outputs $f_{r(k)}$ and $f_{x(k)}$ from the $k$-th layer, which are then passed to the $k+1$-th layer for a similar modeling process.

**Dual-Branch (Asymmetrical).**    Inspired by the prompt learning paradigm (Lester et al., 2021), this architecture models X features as auxiliary prompts (Zhu et al., 2023a; Hong et al., 2024). Specifically, we utilize the HiViT-base backbone (Zhang et al., 2022b) for the RGB branch and adopt the latest modeling approach from OneTracker (Hong et al., 2024) to construct the X branch and enable feature interaction between modalities. The detailed operations in the $k$-th layer of both branches (see Fig. 6 (b)) are as follows:

$$f'_{x(k-1)} = Norm(FFN^{1}_{(k)}(f_{x(k-1)}) + FFN^{2}_{(k)}(f_{r(k-1)})), \tag{24}$$

$$f''_{x(k-1)} = Norm(f_{x(k-1)} + FFN^{3}_{(k)}(f'_{x(k-1)})), \tag{25}$$

$$f_{x(k)} = FFN^{4}_{(k)}(f''_{x(k-1)}), \tag{26}$$

$$f'_{r(k-1)} = Norm(f_{r(k-1)} + \Phi^{RGB}_{CA(k)}(f_{r(k-1)}, f_{r(k-1)})), \tag{27}$$

$$f''_{r(k-1)} = Norm(f'_{r(k-1)} + FFN^{RGB}_{(k)}(f'_{r(k-1)})), \tag{28}$$

$$f_{r(k)} = Norm(f''_{r(k-1)} + f_{x(k)}). \tag{29}$$

In the feature interaction process of the $k$-th layer, the input consists of the output from the previous layer $(k-1)$, *i.e.*, $f_{r(k-1)}$ and $f_{x(k-1)}$. The four fully connected networks $(FFN^{1}_{(k)}, FFN^{2}_{(k)}, FFN^{3}_{(k)}, FFN^{4}_{(k)})$ are used to facilitate inter-modality interaction and update the X modality features, resulting in $f_{x(k)}$. The $k$-th layer of the RGB branch, consisting of an attention network $\Phi^{RGB}_{CA(k)}$ and a feed-forward network $FFN^{RGB}_{(k)}$, is used to process RGB features. Subsequently, $f_{x(k)}$ further influences the RGB modality features (see Eq. 29). Finally, $f_{r(k)}$ and $f_{x(k)}$ are passed to the $k+1$-th layer for similar modeling.

## C.2. Implementation of Spatial Compact Module Variants

As shown in Tab. 6 of Sec. 4.3.1, we conduct ablation studies on the different components of the Spatial Compact Module. Below, we describe the implementation details of each ablation setting.

***w/o* queries.**    This setting removes the modality-specific queries, $q_r$ and $q_r$. Specifically, the initial spatial feature interaction (*i.e.*, Eq. 2, 3, 4, 5) is simplified as:

$$f_r'^t = Norm(f_r^t + \Phi_{CA}(f_r^t, f_x^t)), \tag{30}$$

$$f_x'^t = Norm(f_x^t + \Phi_{CA}(f_x^t, f_r^t)), \tag{31}$$

$$f_r''^t = Norm(f_r'^t + FFN(f_r'^t)), \tag{32}$$

$$f_x''^t = Norm(f_x'^t + FFN(f_x'^t)). \tag{33}$$

Correspondingly, the composition of the compact spatial feature $f_c^t$ also excludes these queries, simplifying Eq. 6 as:

$$f_c^t = f_{c0}^t. \tag{34}$$

**w/o cross-attention.** This setting replaces the bidirectional cross-attention between the two modalities before compact spatial feature generation with self-attention for each modality. Specifically, Eq. 2 and Eq. 3 are modified to:

$$[q_r'; f_r'^t] = Norm([q_r; f_r^t] + \Phi_{CA}([q_r; f_r^t], [q_r; f_r^t])), \tag{35}$$

$$[q_x'; f_x'^t] = Norm([q_x; f_x^t] + \Phi_{CA}([q_x; f_x^t], [q_x; f_x^t])). \tag{36}$$

**unshare embedding.** This setting removes the shared patch embedding and introduces separate patch embeddings for each modality. These patch embeddings share the same network structure.

### C.3. Implementation of Temporal Compact Module Variants

As shown in Tab. 7 of Sec. 4.3.2, we perform ablation studies on different temporal feature representation methods. In this section, we provide the implementation details of each ablation setting.

**w RoI-based.** This setting uses the Region of Interest (RoI) method (Ren et al., 2015) to represent compact temporal features. This approach has been widely adopted in the tracking field, such as in TrDiMP (Wang et al., 2021a) and JointNLT (Zhou et al., 2023). Specifically, we apply RoI processing to the search features $s_c^t$ using the predicted bounding box scaled by 1.5, resulting in the temporal feature $m^t \in \mathbb{R}^{N_m' \times D}$. For fairness, the length of the temporal feature token $N_m'$ obtained by RoI processing is set to be equal to the length $N_m$ of the temporal feature token constructed by our proposed TCM, *i.e.*, $N_m' = N_m = 16$.

**w Query-based.** Recently, several trackers in the visual-only tracking domain have proposed query-based temporal modeling methods, such as ODTrack (Zheng et al., 2024) and AQATrack (Xie et al., 2024). These methods compress the temporal interaction between search and cue features into a small set of temporal queries, providing temporal guidance for the tracker. Since these temporal queries can be seen as a compact way to construct temporal features, we include them in our comparison. Specifically, we define the temporal queries $m_q \in \mathbb{R}^{N_m'' \times D}$ with $N_m'' = N_m = 16$. $m_q$ is transformed into $m^t$ through the following operation:

$$m_q' = Norm(m_q + \Phi_{CA}(m_q, M'^{t-1})), \tag{37}$$

$$m_q'' = Norm(m_q' + \Phi_{CA}(m_q', f_c^t)), \tag{38}$$

$$m^t = Norm(m_q'' + FFN(m_q'')). \tag{39}$$

Here, $M'^{t-1}$ represents the aggregated dense temporal features from $m^i (i \le t-1)$ at different time steps up to $t-1$. The storage and update methods will be described later.

**w $h_i^t$.** This setting selects the top-$N_m$ search tokens from the search features based solely on $h_i^t$, which are then used to represent the compact temporal feature $m^t$.

**w $h_f^t$.** This setting selects the top-$N_m$ search tokens from the search features based solely on $h_f^t$, which are then used to represent the compact temporal feature $m^t$.

*w* $h^t$ ($h_i^t$ **&** $h_f^t$). This setting corresponds to the method adopted by the proposed TCM, which combines $h_i^t$ and $h_f^t$ to construct a refined target distribution heatmap $h^t$, subsequently used to generate the compact temporal feature $m^t$.

The above experimental variants demonstrate different methods of constructing temporal features $m^t$ at a single time step. For storing and updating multi-step temporal features $M^t = \{m^i\}_{i=1}^{L}$, we employ the intuitive and widely used sliding window approach (Xie et al., 2024; Cai et al., 2024) .

For a video sequence with $T$ frames ($1 \leq t \leq T$), the construction of $M^t$ varies at different time steps. At the initial step ($t = 1$), $M^0$ does not yet store temporal features. After obtaining $h_i^1$ from the intermediate tracking results, we first generate the temporal feature $m'^1$ based on $h_i^1$, and initialize the $L$ temporal units in $M^0$. This ensures that the tracker can proceed with forward inference and produce the initial tracking results. Using the final tracking results, we obtain $h_f^1$. Then, we combine it with $h_i^1$ to derive $m^1$, which is used to reinitialize the $L$ temporal units, resulting in $M^1$.

During the time interval $t \in [2, T]$, we use a first-in-first-out sliding window storage method to store the most recent $L$ temporal features. Specifically, we remove the temporal feature with the smallest index and append the newly generated temporal feature unit $m^t$ at the end.

## D. More Qualitative Results

Due to space limitations, Fig. 3 in Sec. 4.2.2 only presents three cases for the qualitative comparison between our model and the latest SOTA models. In this section, we provide additional qualitative comparison results, as illustrated in Fig. 7.

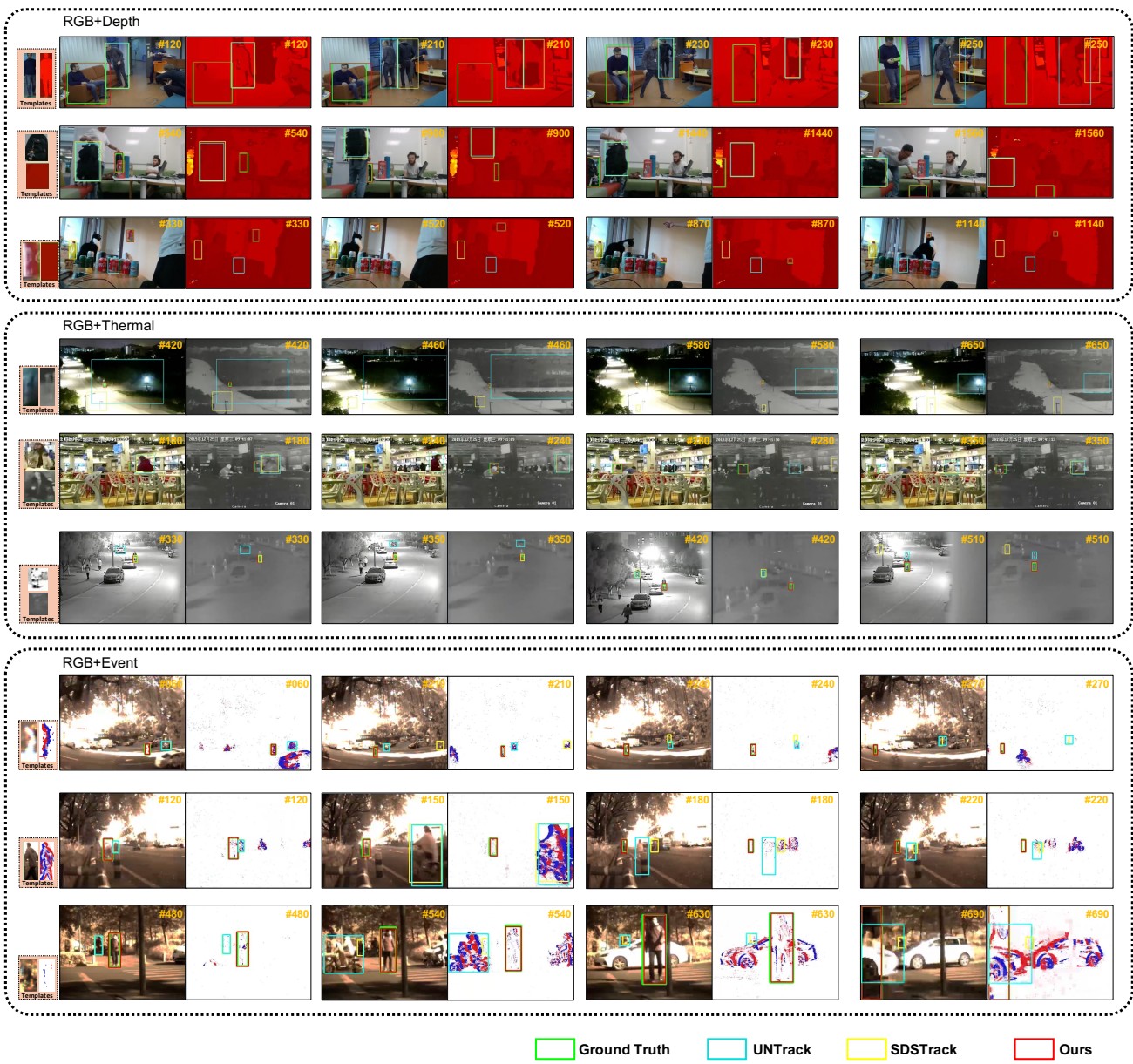

Figure 7: Qualitative comparison results of our tracker with other two trackers (*i.e.*, UNTrack and SDSTrack) on RGB-D/T/E challenging cases. Better viewed in color with zoom-in.

