# OpenReview forum: "CSTrack: Enhancing RGB-X Tracking via Compact Spatiotemporal Features"
_ICML.cc/2025/Conference — ICML 2025 poster_

### Official Review · Reviewer_f47k · 2025-03-08

**Overall Recommendation:** 3

**Summary:**

The article proposes using compact spatiotemporal features for RGB-X tracking. Unlike the commonly used two-stream frameworks, It employs a one-stream structure to reduce computational overhead.  Experiments on multiple downstream tasks demonstrate that this paper achieves the SOTA performance.

## update after rebuttal
The authors' responses have resolved most of my concerns, and I maintain my 'weak accept' rating.

**Claims And Evidence:**

The article is well-written with clear descriptions and motivation, and comprehensive experiments.

**Essential References Not Discussed:**

Some of the latest studies have not been included. For example, TATrack (Temporal adaptive RGBT tracking with modality prompt) and MCTrack (Towards modalities correlation for RGB-T tracking). Additionally, M3PT (Middle fusion and multi-stage, multi-form prompts for robust RGB-T tracking) also attempts to use a single branch to accomplish multimodal tracking tasks, but it differs from the approach presented in this paper. Please compare and discuss these methods.
In addition, it is key to hightlight the difference with other single-stream multi-modal tracking methods, such as:
[1] Unified Single-Stage Transformer Network for Efficient RGB-T Tracking. IJCAI 2024
[2] From Two Stream to One Stream: Efficient RGB-T Tracking via Mutual Prompt Learning and Knowledge Distillation

**Experimental Designs Or Analyses:**

The authors conducted sufficient experiments, including experiments on different modalities and ablation studies on the modules, which basically meet the requirements. However, there are issues with unclear descriptions in the experimental design. For example, it is not clear whether a unified tracker is trained for different modalities or if separate trackers are trained for each modality. Additionally, most of the trackers used for comparison are relatively old, and there is a lack of comparison with the latest trackers.

**Methods And Evaluation Criteria:**

The evaluation metrics and RGB-X tracking datasets are consistent with this paper. However, in the ablation study, it is suggested to modify the dataset for RGB-T experiments to LasHeR, as this dataset is more challenging and authoritative.

**Other Comments Or Suggestions:**

1. The one-stream structure, as a major characteristic of this tracker, is not prominently highlighted, and there is a lack of discussion on other one-stream multimodal trackers, such as ViPT, USTrack, M3PT.  As a distinguishing feature, it is recommended to add content in this regard.
2.  Although the method presented in this paper has a smaller FLOPs, the speed appears to be abnormal, with only 35 fps, which is comparable to that of dual-stream trackers. Is there anything wrong with the experiments?

**Other Strengths And Weaknesses:**

Strengths:
1. This paper proposes a novel approach to integrate fuse spatiotemporal features for multimodal tracking while reducing computational overhead.
2. The experimental results demonstrate the effectiveness of the module proposed in this paper

Weaknesses:
1. some sections could benefit from clearer explanations visualizations.
2. In Table 6, RGBT234 and VisEvent seem to be insensitive to the ablated components, but the paper does not provide an explanation for this observation.
3. The description of SCM is unclear.  In Equations 2, 3, the cross-attention is computed in parallel, while Figure 1 appears to show a serial computation.  Does the order of cross-attention affect the experimental results, or is there an error in Figure 1?

**Questions For Authors:**

1. How does your model differentiate between the inputs of the three modalities?
2. The authors use the HiViT backbone, while most trackers employ the ViT backbone. What are the results if the SCM and TCM are moved to the ViT backbone?
3. The SCM and TCM appear to be transferable to RGB tracking tasks, requiring only modifications to the cross-attention.  Have you conducted such experiments? If so, what is the results?

**Relation To Broader Scientific Literature:**

Previous RGB-X tracking studies typically employed a dual-branch structure and focused on fusion methods. In contrast, this paper uses a one-stream architecture, which is somewhat innovative.

**Theoretical Claims:**

The article focuses on empirical research and validates the effectiveness of CSTrack through experimental results.

---

> ### Author Rebuttal · Authors · 2025-04-01
>
> **Dear Reviewer f47k,**
>
> Thanks for your time and effort in reviewing our work. Your recognition of our novel method, comprehensive experiments, SOTA performance, and writing quality greatly encourages us. We hope the following responses can address your concerns.
> ___
> ### **Q1: Comparison with recent works**
>
> Thank you for mentioning some recent trackers. Here’s how our CSTrack differs:
>
> 1. TATrack and MCTrack use two symmetric backbones for separate RGB and X processing. ViPT employs an asymmetric X branch, while USTrack and Yang [1] concatenate RGB and X feature tokens for multimodal feature integration with a one-stream backbone. However, they still need to process RGB-X dual inputs within the backbone. M3PT uses middle fusion but relies heavily on the early dual-branch architecture for performance.
> 2. In contrast, our CSTrack integrates RGB-X dual inputs into a compact feature space, reducing model complexity and computational costs. Additionally, the table below compares various trackers, highlighting the effectiveness of our method.
> |Tracker|Lasher (SR, PR)|RGBT234 (MSR, MPR)|
> |-|-|-|
> |TATrack|56.1; 70.2|64.4; 87.2|
> |MCTrack|57.1; 71.6|65.6; 87.5|
> |ViPT|52.5; 65.1|61.7; 83.5|
> |USTrack|- |65.8; 87.4|
> |Yang [1]|56.7; 71.4|65.1; 87.3|
> |M3PT|56.1; 70.0|63.9; 86.5|
> |CSTrack-B|60.8; 75.6|70.9; 94.0|
> ___
> ### **Q2: Further explanation of model details**
>
> 1. **Basic model setup.** CSTrack is a unified tracker trained jointly across different datasets, enabling testing without modal differentiation. This is mentioned in Section 4.1 and will be included in the method section of the revised version.
>
> 2. **SCM description.** We apologize for the error in Equation 3 of our paper. The RGB related features should be processed as $[q'_r; f^{'t}_r]$, not $[q_r; f^{t}_r]$. Below is the corrected equation, confirming our use of the serial computation method in Figure 1.
>
>
>     $[q' _x; f^{'t} _x] = Norm( [q _x; f^{t} _x] + \Phi _{CA} ([q _x; f^{t} _x], [q' _r; f^{'t} _r]))$
>
> ___
> ### **Q3: Further explanation of experimental results**
>
> 1. **Ablation study of core modules.** The table below presents ablation results for key modules: Spatial Compact Module (SCM) and Temporal Compact Module (TCM). Adding SCM and TCM improves performance, validating our design. We also offer Lasher benchmark results, consistent with performance rankings observed in RGBT234.
> |Setting|DepthTrack (F; Re; PR)|RGBT234 (MSR; MPR)|Lasher (SR; PR)|VisEvent (SR; PR)|
> |-|-|-|-|-|
> |Baseline|63.6; 64.1; 63.1|68.2; 92.4|57.6; 72.9|62.9; 79.6|
> |+ SCM|65.1; 65.3; 64.9|69.6; 93.0|59.8; 74.7|64.1; 81.0|
> |+ TCM|65.8; 66.4; 65.2|70.9; 94.0|60.8; 75.6|65.2; 82.4|
>
> 2. **SCM analysis.**  Table 6 shows cross-attention and shared embedding in SCM boosts DepthTrack but impacts RGBT234 and VisEvent less. We attribute this to higher noise and fluctuations in depth data compared to thermal and event data, as detailed in Figure 3 and our supplementary videos. This suggests these designs are most effective in challenging scenarios. For RGBT234 and VisEvent,  other designs within SCM are sufficient for strong performance, as shown by SCM's gains across all benchmarks (see the table above).
>
> 3. **Tracking speed.** In Table 5,  our method achieves 35 FPS, surpassing dual-branch trackers at 24 and 18 FPS. This meets expectations and satisfies real-time tracking requirements (exceeding the VOT challenge threshold of 20 FPS).
> ___
> ### **Q4: Additional experimental analysis**
>
> 1. **Performance with ViT backbone.** We retrain CSTrack using the ViT backbone. The results below demonstrate our compact modeling exceeds the performance gains of the HiViT backbone, and our tracker achieves SOTA performance even with the ViT backbone, validating our approach's effectiveness.
> |Backbone|Compact Modeling|DepthTrack (F; Re; PR)|Lasher (SR; PR)|VisEvent (SR; PR)|
> |-|-|-|-|-|
> |HiViT| ✔ |65.8; 66.4; 65.2|60.8; 75.6|65.2; 82.4|
> |HiViT| ✘ |63.6; 64.1; 63.1|57.6; 72.9|62.9; 79.6|
> |ViT| ✔ |63.9; 64.8; 64.3|58.6; 74.3|64.6; 81.1|
>
> 2. **RGB-only tracking test.** Our core compact modeling targets RGB-X dual-stream inputs, making RGB-only tracking not aligned with our objectives. Therefore, we did not conduct this test.
>
> 3. **Impact of bidirectional cross-attention order.** We fine-tune our model by swapping the cross-attention order, and the results show minor performance fluctuations. We speculate that the shared patch embedding and the subsequent one-stream backbone can diminish the influence of interaction order.
> |Setting|DepthTrack (F; Re; PR)|Lasher (SR; PR)|VisEvent (SR; PR)|
> |-|-|-|-|
> |CSTrack-B|65.8; 66.4; 65.2|60.8; 75.6|65.2; 82.4|
> |+ swap order|65.8; 66.0; 65.4|60.7; 75.8|65.2; 82.4|
> ___
> We hope these responses address your concerns and kindly invite you to reassess your rating. Feel free to reach out to us if you have any further questions.
> ___
> [1] From Two Stream to One Stream: Efficient RGB-T Tracking via Mutual Prompt Learning and Knowledge Distillation, Yang et al., in arxiv 2024.

---

### Official Review · Reviewer_W7oh · 2025-03-12

**Overall Recommendation:** 4

**Summary:**

The paper introduces CSTrack, an RGB‐X tracker that leverages a compact spatiotemporal feature representation to improve tracking performance while reducing computational complexity. Unlike existing methods that typically employ dual-branch architectures to process RGB and X modalities separately, CSTrack integrates both modalities into a single compact feature space using bidirectional cross-attention and learnable modality-specific queries. Moreover, it constructs a refined target distribution heatmap by combining intermediate and final tracking results.

**Claims And Evidence:**

Yes.

**Essential References Not Discussed:**

Not to my knowledge.

**Experimental Designs Or Analyses:**

Yes, the experimental design leverages foundational benchmarks in the relevant domain.

**Methods And Evaluation Criteria:**

Yes.

**Other Comments Or Suggestions:**

1. The method for modality missing simply duplicates the available modality data (e.g., copying RGB to the X channel) without accounting for potential noise interference in the X modality. This approach may inadvertently introduce erroneous signals when the auxiliary modality is noisy.
2. The paper uses a joint training strategy with both RGB‐X and RGB datasets, but lacks a detailed analysis of their distribution differences and potential domain shift issues.

**Other Strengths And Weaknesses:**

1. Simplifying the dual-stream network in RGB‐X tracking by integrating both modalities into a single compact feature space is a sound idea, as it helps reduce computational complexity and simplifies the model architecture.
2. Constructing a refined target distribution heatmap by combining intermediate and final tracking results is an effective practical method for selecting key target features, thereby enhancing tracking robustness.

**Questions For Authors:**

1. For the Spatial Compact Module, could you elaborate on how the bidirectional cross-attention and modality-specific queries behave when one modality (e.g., thermal or depth) provides low-quality or noisy data?

**Relation To Broader Scientific Literature:**

Traditionally, many methods—such as ViPT, OneTracker, and SDSTrack—rely on dual-branch architectures to process RGB and auxiliary modalities separately. CSTrack builds on these approaches by unifying the two modalities into a single compact feature space, a strategy that mirrors recent trends in transformer-based models where learnable queries and cross-attention mechanisms have proven effective for capturing complex relationships in visual data.

**Theoretical Claims:**

The paper does not include formal proofs for its theoretical claims. Instead, its contributions are presented through the design of SCM and TCM modules, which are supported by extensive empirical evaluations and ablation studies.

---

> ### Author Rebuttal · Authors · 2025-04-01
>
> **Dear Reviewer W7oh,**
>
> We sincerely appreciate your thorough review of our work and are grateful for your recognition of our sound motivation, effective method, and extensive experimental analysis. In response to your concerns, we have provided detailed explanations below:
> ___
> ### **Q1: Noise Interference in modality-missing tracking**
>
> Thanks for your constructive suggestion on analyzing noise impact in modality-missing tracking task. IPL [1] recently proposes the modality-missing tracking benchmarks, along with a baseline method that duplicates available modality data, which we adopt. As shown in Tables 2 and 4 of the paper, modality-missing settings significantly degrade model performance. We attribute this mainly to the influence of two types of noise:
>
> 1. **Miss of advantage modality.** Figure 4 in the paper introduces the concept of the advantage modality, which primarily conveys object appearance information in challenging scenarios. In modality-missing task, this advantage modality may be unavailable, leaving the tracker susceptible to noise interference from the non-advantage modality due to its lack of appearance features.
>
> 2. **Training and inference bias.** CSTrack is trained with complete RGB-X modalities but directly inferred in modality-missing settings. The difference in input data types introduces noise that significantly affects performance. Despite this, CSTrack exceeds IPL, a tracker tailored for such scenarios. We think this is primarily due to our shared patch embedding, which transforms initial RGB and X data into a unified feature space, improving compatibility with varied inputs. Below, we test the impact of shared embedding to further confirm our analysis:
> |Setting | LasHeR-Miss (SR; PR) | RGBT234-Miss (MSR; MPR) |
> |-|-|-|
> |Unshared Embedding | 48.7, 60.6 | 65.3, 86.5 |
> |Shared Embedding | 50.3, 62.2 | 68.0, 89.8 |
> ___
> ### **Q2: Distribution analysis of RGB and RGB-X datasets**
>
> We appreciate your valuable suggestions on the distribution analysis of various modality datasets. The analysis is as follows:
>
> 1. **Distribution differences across modality datasets.** We perform statistical analyses on representative RGB and RGB-D/T/E datasets to obtain normalization mean values of training videos, including the ImageNet for comparison as a reference for natural image distribution:
> |Datasets| ImageNet | LaSOT (RGB-only) |DepthTrack (RGB-D)|Lasher (RGB-T)|VisEvent (RGB-E)|
> |:-|:-|:-|:-|:-|:-|
> | RGB Mean| 0.485; 0.456; 0.406 | 0.456; 0.459; 0.426 | 0.417; 0.414; 0.393 | 0.500; 0.499; 0.471 | 0.418; 0.375; 0.317 |
> | X Mean| - | -| 0.574; 0.456; 0.240 | 0.372; 0.372; 0.368 | 0.935; 0.904; 0.949 |
>
>     - RGB modality: LaSOT and DepthTrack, collected in typical natural environments, have mean values similar to ImageNet. However, Lasher, often focusing on dark environments, and VisEvent, which collects predominantly yellow RGB images, deviate from these natural image means.
>
>     - X modality: as shown in Figure 3 of the paper, X modalities vary greatly in image style, resulting in significant differences in mean values.
>
> 2. **Model's discriminative ability across modality datasets.** Using features from the one-stream backbone (see Equation 8 in the paper), we extract tokens associated with learnable queries and build a four-class classifier with two fully connected layers to identify the input modality type: RGB-only, RGB-D, RGB-T, and RGB-E. We freeze the existing model and train this classification head for 2 epochs. The accuracy results show the model can effectively distinguish different modality datasets, thanks to the clear distribution differences.
> |Datasets|LaSOT (RGB-only)|DepthTrack (RGB-D)|Lasher (RGB-T)|VisEvent (RGB-E)|
> |:-|:-|:-|:-|:-|
> |Accuracy|100%|98%|100%|100%|
> 3. **Advantages of joint training.** Our ablation study, detailed in Table 8 (#3) of the paper, shows that the benefits from joint training, including larger datasets and knowledge sharing across modalities, outweigh the drawbacks of domain shift, resulting in enhanced model performance.
> ___
> ### **Q3: Function of bidirectional cross-attention and modality-specific queries.**
>
> Our spatial compact module aims to integrate RGB and X input streams into a unified feature space, facilitating simplified and effective spatial modeling. When a modality has low-quality or noisy data, termed "non-advantage modality" (see Figure 4), bidirectional cross-attention first emphasizes the advantage modality's representations while minimizing those of the non-advantage modality. Next, modality-specific queries preserve global semantic information of each modality, offering additional reference information for subsequent feature integration.
> ___
> We hope these explanations can resolve your concerns. If you have any more questions or need further clarification, feel free to reach out to us.
> ___
> [1] Modality-missing RGBT Tracking: Invertible Prompt Learning and High-quality Benchmarks,  Lu et al., in IJCV 2O24.

---

### Official Review · Reviewer_waVm · 2025-03-13

**Overall Recommendation:** 2

**Summary:**

This paper introduces CSTrack, a novel RGB-X tracker designed to enhance tracking performance by leveraging compact spatiotemporal features. Traditional RGB-X trackers typically process RGB and auxiliary modality (X) inputs separately using dual-branch architectures, which increases computational complexity and limits effective feature fusion. To address this limitation, CSTrack proposes a single-branch compact feature representation that integrates spatial and temporal information more efficiently. This approach incorporates two key modules: the Spatial Compact Module (SCM) and the Temporal Compact Module (TCM). The method is evaluated across several benchmarks, including RGB-D, RGB-T, and RGB-Event tracking datasets, such as DepthTrack, VOT-RGBD2022, LasHeR, RGBT234, and VisEvent.

**Claims And Evidence:**

Overall, the paper presents well-supported claims with convincing evidence through extensive experiments and ablation studies.

**Essential References Not Discussed:**

No.

**Experimental Designs Or Analyses:**

The experimental design in the paper is well-structured and comprehensive, evaluating CSTrack across multiple datasets and conducting ablation studies to validate its core components.

**Methods And Evaluation Criteria:**

Yes.

**Other Comments Or Suggestions:**

No.

**Other Strengths And Weaknesses:**

Strengths:
1. The paper is well-structured and easy to follow. Extensive supplementary material provides additional experimental insights, dataset details, and qualitative comparisons.
2. The Spatial Compact Module (SCM) and Temporal Compact Module (TCM) offer a novel approach to multimodal feature integration while reducing computational overhead.
3. The use of compact spatiotemporal representations is an innovative departure from the traditional dual-branch RGB-X tracking architectures.


Weaknesses:
1. The proposed method does not achieve state-of-the-art performance on some of the metrics compared to those in the paper 'Exploiting Multimodal Spatial-Temporal Patterns for Video Object Tracking' (AAAI 2025).
2. The reduction in FLOPs compared to dual-branch methods is not significant (only a minor drop from 38G to 36G) in Table 5.
3. It is recommended to compare the efficiency of the proposed method, including parameters and FLOPs, with other methods to better highlight its computational performance.

**Questions For Authors:**

No.

**Relation To Broader Scientific Literature:**

RGB-X Tracking and Multimodal Fusion: Traditional RGB-X tracking methods typically use dual-branch architectures,
Temporal Feature Modeling in Tracking:  Many existing RGB-X trackers lack effective temporal modeling.

**Theoretical Claims:**

The paper primarily presents a methodological and empirical contribution rather than a theoretical one.

---

> ### Author Rebuttal · Authors · 2025-04-01
>
> **Dear Reviewer waVm,**
>
> Thank you for your time and effort in reviewing our work. We appreciate your recognition of our novel, innovative, and efficient method, along with the comprehensive, well-structured, and convincing experiments, as well as easy-to-follow writing and the extensive supplementary material. We note your concerns about the performance and computational efficiency of our model. In response, we've conducted targeted comparative analyses and additional experiments, aimed at addressing your concerns.
> ___
> ### **Q1: Performance comparison with STTrack [1]**
>
> We provide a detailed comparison between CSTrack and STTrack across various metrics:
> | Model | DepthTrack (F; Re; PR) | VOT-RGBD22 (EAO; Acc; Rob) | Lasher (SR; PR) | RGBT234 (MSR; MPR) | VisEvent (SR; PR) | Params | Flops |
> |-|-|-|-|-|-|-|-|
> | STTrack | 63.3; 63.4; 63.2 | 77.6; 82.5; 93.7 | 60.3; 76.0 | 66.7; 89.8 | 61.9; 78.6 | 128M | 91G |
> | CSTrack-B | 65.8; 66.4; 65.2 | 77.4; 83.3; 92.9 | 60.8; 75.6 | 70.9; 94.0 | 65.2; 82.4 | 75M | 36G |
> | CSTrack-L | 67.1; 67.5; 67.3 | 78.2;83.8;93.2 | 61.8; 77.1 | 71.6; 96.0 | 66.1; 82.9 | 254M | 110G |
>
>
> 1. **CSTrack-B, the model presented in our paper, exhibits superior average performance.** As noted, CSTrack-B exhibits performance comparable to STTrack in VOT-RGBD22 (EAO ↓ 0.2%; Acc ↑ 0.8%; Rob ↓ 0.8%) and Lasher (SR ↑ 0.5%; PR ↓ 0.4%). However, CSTrack-B significantly outperforms STTrack in other benchmarks, achieving improvements such as PR ↑ 2.0% in DepthTrack, MPR ↑ 4.2% in RGBT234, and PR ↑ 3.8% in VisEvent, contributing to a superior average performance.
>
> 2. **CSTrack targets a more challenging optimization objective.**
> Unlike our CSTrack, which uses a single model weight to simultaneously handle RGB-D/T/E tasks, STTrack optimizes separate weights for each task. Additionally, the STTrack repository suggests that it even optimizes individual checkpoints for each benchmark. While UnTrack [2] indicates that this approach may yield better performance, it is a less practical strategy for real-world applications.
>
> 3. **CSTrack-B offers superior computational efficiency.**
> CSTrack-B offers a substantial advantage in Params (↓ 53M) and Flops (↓ 55G), achieving a better performance-computation balance. Focusing on maximizing performance, we develop CSTrack-L by adopting a larger backbone, surpassing STTrack across all benchmarks.
>
> ___
> ### **Q2: FLOPs reduction compared to dual-branch methods**
>
> As shown in Table 5 of the paper, our single-branch method reduces FLOPs by only 2G compared to asymmetrical dual-branch methods. Here's the explanation:
>
> 1. Asymmetrical dual-branch methods often incorporate parameter-intensive prompter modules into each layer of the RGB branch to integrate RGB-X features, as seen in the fully connected networks in OneTracker [4]. This results in some FLOPs benefits but leads to increased parameters (↑ 9M) and speed limitations (↓ 11 FPS).
>
> 2. Table 5 indicates a notable disadvantage in tracking performance with this dual-branch method, e.g., MSR ↓ 3.7% in RGBT234. It suggests that the scale of FLOPs in this method makes it difficult to effectively integrate RGB-X features.
> ___
> ### **Q3:  Computational efficiency comparison with existing methods**
>
> Thanks for your constructive suggestion.  We analyze the computational efficiency of recent open-source RGB-X methods, yielding the following results:
>
> |Model|DepthTrack (F; Re; PR) | RGBT234 (MSR; MPR) | VisEvent (SR; PR) | Params | Flops | Speed |
> |-|-|-|-|-|-|-|
> |UnTrack [2] | 61.0, 61.0, 61.0 | 62.5, 84.2 | 58.9, 75.5 | 99M | 24G | 24 FPS|
> |SDSTrack [3] | 61.4, 60.9, 61.9 | 62.5, 84.8 | 59.7, 76.7 | 102M | 108G | 22 FPS|
> |STTrack [1] | 63.3, 63.4, 63.2 | 66.7, 89.8 | 61.9, 78.6 | 128M | 91G | 27 FPS|
> |CSTrack-B | 65.8, 66.4, 65.2 | 70.9, 94.0 | 65.2, 82.4 | 75M | 36G | 33 FPS|
>
> CSTrack-B excels in Params and Speed, with only Flops lagging behind UnTrack. Similar to the analysis in **Q2**, UnTrack's core modules primarily consist of fully connected networks, offering Flops advantages but limits Params, Speed, and tracking performance. Overall, our method demonstrates superior tracking performance and computational efficiency.
> ___
> We hope these responses address your concerns and would be grateful if you could reconsider your rating. Should you have any additional feedback or require further clarification, please do not hesitate to let us know.
> ___
> [1] Exploiting Multimodal Spatial-Temporal Patterns for Video Object Tracking, Hu et al., in AAAI 2025.
>
> [2] Single-Model and Any-Modality for Video Object Tracking, Wu et al., in CVPR 2024.
>
> [3] SDSTrack: Self-Distillation Symmetric Adapter Learning for Multi-Modal Visual Object Tracking, Hou et al., in CVPR 2024.
>
> [4] OneTracker: Unifying Visual Object Tracking with Foundation Models and Efficient Tuning, Hong et al., in CVPR 2024.

---

> > ### Comment · Reviewer_waVm · 2025-04-04
> >
> > Thank you for your response. Regarding the first question, CSTrack-L outperforms STTrack; however, STTrack serves as a base model with a search resolution of 256 and a template size of 128. Therefore, the comparison may not be entirely fair.

---

> > > ### Author Response · Authors · 2025-04-05
> > >
> > > **Dear Reviewer waVm,**
> > >
> > > Thank you for your timely feedback. Regarding your remaining concern about **Q1 (Performance comparison with STTrack)**, we would like to provide further clarification.
> > >
> > > ___
> > > ### **Clarification 1: CSTrack and STTrack utilize the same resolution for search and template images.**
> > >
> > > It is important to note that CSTrack-L is merely an enhanced version of CSTrack-B, achieved solely by employing a larger backbone. Therefore, CSTrack-L, CSTrack-B, and STTrack all utilize a search image resolution of 256 and a template resolution of 128, ensuring fairness concerning input image resolution. (Due to word limits in the comments window, we did not provide a detailed explanation of the experimental setup for CSTrack-L. We apologize for any confusion this might have caused.)
> > >
> > > ___
> > > ### **Clarification 2: When CSTrack-B adopts the same optimization objective as STTrack, it outperforms STTrack across all datasets.**
> > >
> > > As discussed in the initial rebuttal regarding Q1, our CSTrack-B employs an optimization objective based on a single unified
> > >  model weight to simultaneously handle RGB-D/T/E tasks. In contrast, STTrack adopts the approach of optimizing a separate set of model weights for each modality, which simplifies the complexity of the optimization target. As demonstrated in UnTrack, this strategy often results in superior performance. For a fair comparison, we also evaluated CSTrack-B using modality-specific optimization weights. As shown in the table below, modality-specific CSTrack-B surpasses STTrack across all metrics while offering significant computational efficiency advantages.
> > >
> > > | Model | DepthTrack (F; Re; PR) | VOT-RGBD22 (EAO; Acc; Rob) | Lasher (SR; PR) | RGBT234 (MSR; MPR) | VisEvent (SR;PR) | Params | Flops |
> > > |-|-|-|-|-|-|-|-|
> > > | STTrack | 63.3; 63.4; 63.2 | 77.6; 82.5; 93.7 | 60.3; 76.0 | 66.7; 89.8 | 61.9; 78.6 | 128M | 91G |
> > > | CSTrack-B (unified) | 65.8; 66.4; 65.2 | 77.4; 83.3; 92.9 | 60.8; 75.6 | 70.9; 94.0 | 65.2; 82.4 | 75M | 36G |
> > > | CSTrack-B (modality-specific) | 65.9; 66.8; 65.5 | 77.8; 83.6; 93.8 | 60.9; 76.4 | 70.9; 94.4 | 65.4; 82.5 | 75M | 36G |
> > >
> > > ___
> > > We believe these additional clarifications have addressed your concerns and would greatly appreciate it if you could reconsider your rating, as it is highly important to us.
> > >
> > > Due to the limitations on the number of comments, if there are any other issues you would like us to clarify, please update them in the **'Rebuttal Comment'** window above. We will promptly address them and provide updates in this window. Our sole purpose in doing this is to sincerely thank you again for your recognition of our method, experiments, and writing in the initial review, and we genuinely hope to address any remaining doubts you might have about our work.
> > >
> > > Wishing you all the best.

---

### Decision · Program_Chairs · 2025-05-01

**Decision:**

Accept (poster)

**Comment:**

Based on the reviewers’ feedback and the authors’ rebuttals, I recommend weakly accepting this paper. The submission presents a novel approach to RGB-X tracking by integrating multimodal data into a compact feature space, which simplifies the model architecture and reduces computational complexity. The extensive experiments and ablation studies provide strong empirical support for the proposed method. While there are some limitations, such as the handling of noisy data and the need for further analysis on dataset distributions, the authors have addressed these concerns adequately in their rebuttal. The innovations and performance improvements demonstrated by CSTrack make it a valuable contribution to the field of multimodal tracking.